# A new class of binding-protein dependent solute transporter exemplified by the TAXI-GltS system from *Bordetella pertussis*
Lily M. Jaques[1], Joseph F. S. Davies[1], Jack J. Sheldon-Towler[1], David J. Kelly [2], Vanessa Leone [3] & Christopher Mulligan [1] ✉

Tripartite ATP-independent periplasmic (TRAP) transporters are widespread in prokaryotes, but absent in eukaryotes, and transport various substrates. TRAP transporters are typically composed of a monomeric substrate binding protein (SBP) and a characteristic transmembrane component. Here, we describe the discovery and characterisation of a TRAP SBP from the TAXI subfamily with a previously unidentified architecture. BP0403 from *Bordetella pertussis* is a predicted lipoprotein with 3 distinct domains; an α/β globular domain, a helical domain and a C-terminal TAXI SBP domain. Characterisation of full-length BP0403 reveals that it forms a stable dimer, and structural modelling coupled with molecular weight analysis reveals that the interdomain helical region is solely responsible for dimerisation. Differential scanning fluorimetry (DSF) and intrinsic tyrosine fluorescence reveal that BP0403 binds L-glutamate with nanomolar affinity. Unexpectedly, genome context analysis of BP0403 reveals no TRAP membrane component genes; instead, we find co-localisation and translational coupling with *gltS*, encoding a Na$^+$/glutamate symporter. In other bacteria, we identified fused BP0403-GltS homologues, strongly suggesting that this constitutes a completely novel SBP-dependent secondary active transporter. Structural comparisons suggest GltS operates by an elevator-type mechanism, like TRAP transporters; the association of an SBP with this class of secondary transporter is an emerging theme.

Substrate binding proteins (SBPs) are obligate components of many solute transporters in prokaryotes, where they deliver solutes to integral membrane transporters, often conferring stringent selectivity, high affinity, and directionality to solute uptake. SBPs have been recruited by three main transporter families; the primary-active ATP-binding cassette (ABC) transporters that hydrolyse ATP to power transport[1], and the secondary-active tripartite tricarboxylate transporter (TTT) and tripartite ATP-independent periplasmic (TRAP) transporter families, both of which harness electrochemical ion-gradients to drive transport[2–4].

TRAP transporters (TCDB ID: 2.A.56) are the largest and best characterised SBP-dependent secondary active transporter family and contribute to virulence of several human pathogens[4]. TRAP transporters are composed of an SBP and two unequally sized membrane components that can exist as two separate proteins or fused into a single polypeptide[2,5]. The

TRAP family can be further divided into two subfamilies; the DctP-type and the TAXI (TRAP associated extracytoplasmic immunogenic) TRAP transporters; both subfamilies have homologous membrane components, but the SBPs share no amino acid sequence similarity[2]. DctP-type TRAPs were the first discovered and are named after the first TRAP transporter identified, DctPQM from *Rhodobacter capsulatus*, which is C4-dicarboxylate specific[5]. DctP-type TRAP systems have been extensively characterised, revealing a diverse portfolio of substrates, most commonly aliphatic or aromatic organic acids[6–9].

Recent structural characterisation of fused DctP-type TRAP membrane components reveal that they are monomeric proteins composed of two domains, a scaffold domain and a transport domain, the latter of which houses all substrate binding residues[10–12]. TRAP membrane components share the same fold as members of the divalent anion Na$^+$ symporter

[1]School of Biosciences, Division of Natural Sciences, University of Kent, Canterbury, UK. [2]School of Biosciences, The University of Sheffield, Sheffield, UK. [3]Department of Biophysics and Data Science Institute, Medical College of Wisconsin, Milwaukee, WI, USA. ✉e-mail: c.mulligan@kent.ac.uk

(DASS) family[13–15]. As such, they also share a general transport mechanism in which the transport domain undergoes an elevator-like movement through the membrane to facilitate transport[10–12,16,17]. Similarly to DASS family members[18], liposome reconstitution of DctP-type TRAP systems has revealed that transport is dependent on the presence of a $Na^+$ gradient[11,12,19,20].

The TAXI TRAP subfamily is poorly characterised in comparison, with only a handful of substrates known or postulated and only two experimentally determined structures currently published[21,22]. The TAXI TRAP systems differ in several ways from their DctP-type cousins. Whereas DctP-type TRAP transporters can have either fused or separate membrane components, TAXI systems invariably have a fused membrane protein[2]. In addition, while both DctP-type and TAXI TRAP systems are widespread in bacteria, the TAXI group are the *only* TRAP systems found in archaea. In contrast to the $Na^+$-dependence of DctP-type transporters, at least one TAXI TRAP transporter harnesses the proton-motive force to drive transport[23]. In addition to these differences, a higher proportion of TAXI SBPs appear to be lipoproteins compared to DctP-type TRAP SBPs, although the significance of this is not clear[23].

Although DctP-type and TAXI SBPs are unrelated at the sequence level, their domain architecture and overall structures are similar. They are usually monomeric, 300–350 amino acids long and are invariably solely composed of 2 α/β globular domains and a hinge region, which closes upon interaction with substrate in a "venus flytrap"-like mechanism[24,25]. While there are examples of 'orphan' TRAP SBPs[2], the vast majority of the genes encoding them are operonic, with the genes encoding the characteristic TRAP membrane proteins with which they interact. However, in this work we describe the discovery and characterisation of a novel elongated, dimeric TAXI SBP from the human pathogen *Bordetella pertussis*, which has recruited an additional globular domain. We present compelling evidence that this unusual TAXI SBP binds glutamate but is not part of a TRAP transporter; rather, it utilises a member of the unrelated GltS family of $Na^+$ coupled glutamate symporters for uptake. This conclusion is strengthened by the identification of TAXI-GltS fusion proteins encoded in the genomes of other bacteria. This unprecedented architecture reveals for the first time the existence of a novel, previously uncharacterised type of SBP-dependent transporter.

## Methods
### Molecular biology
Genes encoding full-length BP0403 (BP0403$_{FL}$, Uniprot ID: Q7VSK8), BP0403 N-terminal domain (BP0403$_{NTD}$) and BP0403 C-terminal domain (BP0403$_{SBP}$) were codon optimised, synthesised and cloned into pET151 by Invitrogen GeneArt. For both BP0403$_{FL}$ and BP0403$_{NTD}$, the N-terminal signal peptide (MTMFIRWLILSACLLLAAC) was not included in the final construct.

### Protein expression and purification
To express BP0403$_{FL}$, BP0403$_{NTD}$ and BP0403$_{SBP}$, *E. coli* BL21 (DE3) was transformed with a pET151 plasmid with the gene encoding these proteins in frame with an N-terminal His$_6$ affinity purification tag. The expression strains were grown in LB media supplemented with 100 µg/ml ampicillin in a 2.5 L baffled Tunair flask. The cultures were grown at 37 °C to an OD600 of 0.8, at which point expression was induced by addition of 1 mM IPTG. Cultures were incubated for a further two hours, then harvested by centrifugation at 4000 rcf for 20 min and resuspended in Lysis buffer (50 mM Tris-HCl, pH 8, 200 mM NaCl, 5% glycerol). Cells were lysed by ultrasonication and the lysate was clarified by centrifugation at 20,000 rcf for 20 min at 4 °C.

The clarified lysate was applied to Ni-NTA resin (Qiagen) at room temperature. The resin was washed with 20 column volumes (CV) of Wash Buffer (50 mM Tris-HCl, pH 8, 100 mM NaCl, 5% glycerol, 20 mM imidazole), and the bound protein was eluted with 5 × 0.5 CV of Elution Buffer (50 mM Tris-HCl, pH 8, 100 mM NaCl, 5% glycerol, 300 mM imidazole). The protein was further purified using size exclusion chromatography with a Superdex 200 Increase 10/300 GL column (GE Healthcare) equilibrated with SEC buffer (50 mM Tris-HCl, pH 8, 100 mM NaCl, 5% glycerol).

### SEC-based oligomeric state analysis
For oligomeric state analysis, IMAC purification elution fractions were pooled and concentrated, incubated at room temperature for 20 min with 5 mM EDTA and 100 mM glycerol, and applied to a Superdex 200 Increase 10/300 GL column equilibrated with SEC buffer. SEC was performed at a flow rate of 0.5 ml/min with SEC buffer. The calibration curve was generated by determining the elution volumes of SEC standards (Thermo Scientific) applied to the same column under the same conditions and plotting the elution volumes as a function of log molecular weight of the standards.

### Differential Scanning Fluorimetry (DSF)
DSF was performed by mixing the protein of interest, either 1.5 µM BP0403$_{FL}$, 19 µM BP0403$_{SBP}$, or 6.5 µM BP0403$_{NTD}$, with 1x SYPRO Orange dye (Merck) in DSF buffer (50 mM Tris-HCl, pH 8, 20 mM NaCl). Potential ligands were added to a final concentration of 1 mM, where appropriate. A QuantStudio 3 RT-PCR thermocycler (Invitrogen) was used to initially cool to 5 °C at a rate of 1.6 °C/s. The temperature was then increased to 95 °C at 1.6 °C/s and readings were taken every 1 °C while holding the temperature for 5 s using the SYBR reporter dye setting. DSF data was exported to Microsoft Excel and GraphPad Prism for analysis and presentation.

### Intrinsic tyrosine fluorescence
Fluorescence assays was measured using a Cary Eclipse Fluorimeter (Aglient) in a 2 ml quartz cuvette. Ligand titrations were performed using 50 nM SEC-purified BP0403$_{SBP}$ in 50 mM Tris, pH 7.4 in time-based acquisition mode with excitation and emission wavelengths of 280 and 310 nm, respectively. Excitation slit width was set to 5 nm, emission slit was set to 10 nm, the photomultiplier tube (PMT) voltage was set to 950 V. Emission spectra were collected using 500 nM SEC-purified BP0403$_{SBP}$ with an excitation wavelength of 280 nm, and an emission range of 280–400 nm. Excitation slit width was set to 5 nm, emission slit was set to 10 nm, the photomultiplier tube (PMT) voltage was set to 800 V, and a Savitzky-Golay smoothing factor of 15 was applied. Binding curves were analysed using GraphPad Prism and $K_d$ values were obtained by fitting the curves to a one site specific binding model:

$$Y = B\max \times X/(Kd + X)$$

### Protein modelling
Structural models of the BP0403:glutamate complex, the BP0403 dimer, GltS dimer, and the BP0403:GltS heterotetramer were built using AlphaFold 3[26]. The sequences of BP0403 (accession ID: Q7VSK8) and GltS (accession ID: Q7VSK9) of *B. pertussis* were taken from the UniProt database. For the glutamate bound to the BP0403, we performed 250 predictions. For BP0403 dimer and BP0403: GltS tetramer, we made 1500 predictions each and for the GltS dimer we generated 5000 model. We selected the one with the highest combined predicted Template Model (pTM) and interface pTM score. Images were generated using UCSF ChimeraX[27].

### Statistics and reproducibility
Details of statistics and reproducibility are provided in the figure legends in which the corresponding data are presented. Where replicates were performed, the number of replicate (n), the average and the standard error of the mean (SEM) is reported.

## Results
### Elongated TAXI SBPs are predicted to have multiple domains
In our previous analysis of the distribution and variation of TAXI SBPs[22], we observed that one SBP, BP0403 from *Bordetella pertussis*, was substantially larger than the other TAXI proteins in our sequence alignment. While all other TAXIs SBPs in the alignment were 300-350 amino acids long, similar to DctP-type SBPs[2], BP0403 was composed of 503 amino acids. Intrigued by

this surprisingly long TAXI SBP, we investigated its predicted structure using the AlphaFold Protein Structure Database[28].

The BP0403 model structure revealed a large globular protein that could be divided into 3 distinct domains (Fig. 1A); an N-terminal α/β globular domain (NTD), a long helical domain (HD), and a large C-terminal α/β globular domain composed of a standard SBP architecture (SBP domain). To confirm that the C-terminal domain indeed has an SBP fold, we aligned the predicted structure of the BP0403 SBP domain with the known structure of the glutamate-specific TAXI SBP, VcGluP[22], which superimposed with an RMSD of 3.4 Å across 293 residues, confirming its identity (Fig. 1B). As all other SBPs are secreted into the periplasm or are lipid-anchored periplasmic-facing proteins[2], we analysed the BP0403 amino acid sequence using SignalP 6.0[29], which revealed the presence of an N-terminal lipoprotein signal peptide (Sec/SPII) that would tether the protein to the outer leaflet of the inner membrane (Fig. 1A).

To shed light on the potential domain arrangement in BP0403, we analysed the AlphaFold2 model of the BP0403 homologue BPP3828 from *Bordetella parapertussis*, which modelled the helical domain in a more extended arrangement (Fig. 1C). This arrangement suggests a possible mechanism in which the SBP is tethered to the membrane but has substantial freedom of movement due to the flexible helical domain.

### BP0403 binds L-glutamate

To investigate the structural and functional characteristics of BP0403, we expressed the gene in *E. coli* with an N-terminal histidine tag in place of the lipoprotein signal peptide. Overexpressed BP0403 was purified using IMAC, which revealed an intense band on SDS-PAGE at ~50–55 kDa, consistent with the molecular weight of 53.7 kDa estimated from the amino acid sequence (Fig. 2A, inset). Further purification using size exclusion chromatography (SEC) revealed a single symmetrical peak, indicative of folded and stable protein (Fig. 2A).

To investigate the ligand specificity of BP0403 we used differential scanning fluorimetry (DSF), which provides a readout of the melting temperature (Tm) of the protein, which often increases upon ligand binding, due to increased protein stability[30]. DSF analysis of conventional TAXI SBPs consistently produces a single trough, indicative of a single melting event, for example, with the L-glutamate specific TAXI SBP, VcGluP (Fig. 2B, *inset*)[22]. However, analysis of BP0403$_{FL}$ using DSF revealed an unusual double dip pattern (Fig. 2B), suggesting the occurrence of two melting events with separable Tms of 43.7 °C (Tm$_1$) and 53.7 °C (Tm$_2$), which we reasoned likely reflects the independent melting of the globular SBP and N-terminal domains.

To identify the ligand binding specificity of BP0403$_{FL}$, we screened a library of 14 different compounds consisting of a range of organic anions, with similar features to "classical" TRAP ligands, using DSF (Supplementary Fig. 2). While the majority of the compounds induced no change in the position of either of the melt troughs of BP0403$_{FL}$ (Fig. 2C), we observed a pronounced rightward shift of only Tm$_2$ in the presence of L-glutamate (Fig. 2D). These data strongly support the conclusion that Tm$_2$ relates to the SBP domain and identifies L-glutamate as the ligand.

To determine which melt trough belonged to which globular domain, we performed DSF on the isolated domains, BP0403$_{NTD}$ (residues 20–139)

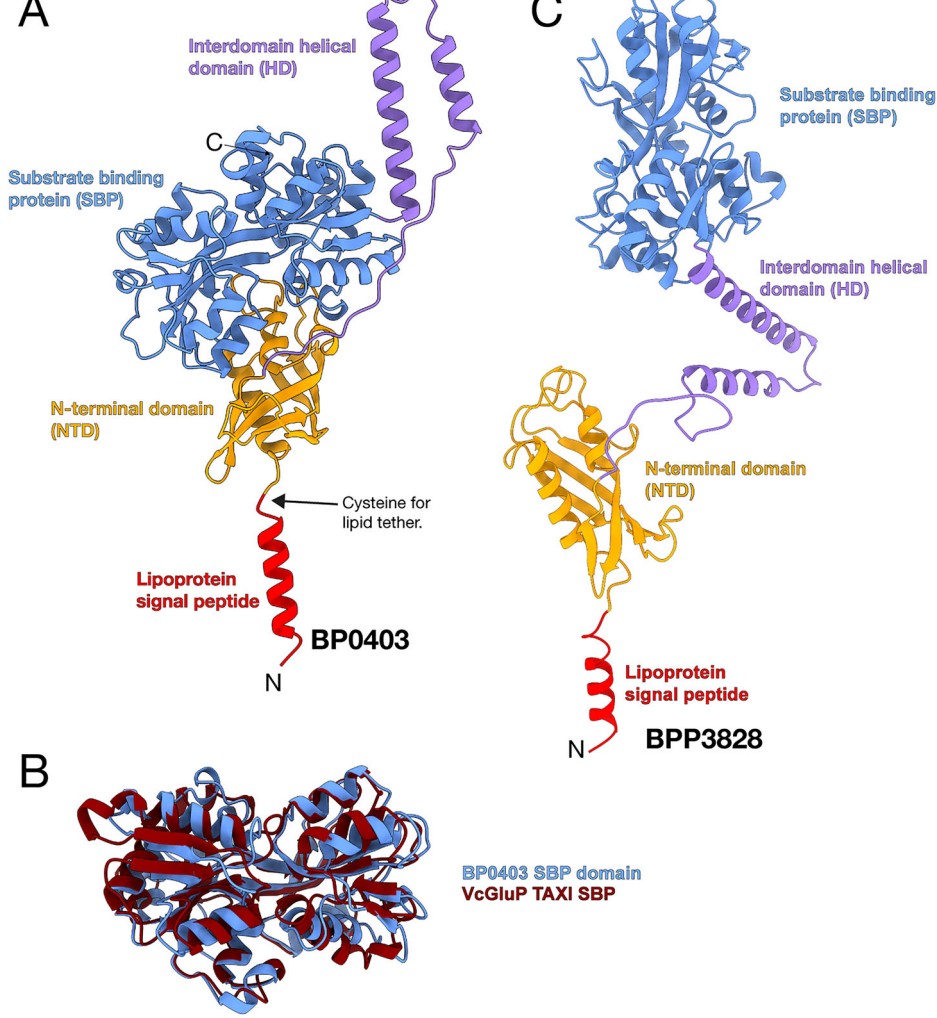

**Fig. 1 | Structural model of BP0403. A** Model of BP0403 from the AlphaFold Protein Structure Database colour coded as follows; lipoprotein signal peptide (red), N-terminal domain (orange), interdomain helical region (purple), and SBP domain (blue, LDDT representation is in Supplementary Fig. 1) **B** Superimposition of the SBP domain from BP0403 and the crystal structure of glutamate-specific TAXI, VcGluP (PDB ID: 8S4J). **C** Model of BPP3828 from *B. parapertussis* from the AlphaFold Protein Structure Database revealing an elongated structure (same colour code as in **A**).

**Fig. 2 | Purification and differential scanning fluorimetry (DSF) analysis of BP0403. A** Size exclusion chromatography (SEC) trace of IMAC purified BP0403. *Inset*: SDS-PAGE of IMAC fractions from BP0403 purification, including molecular weight ladder (L), flowthrough (FT), the wash fraction (W), and elution fractions 1-5. **B–D** Derivatives of the unfolding curves (dF/dT) for (**B**) BP0403 in the absence of ligand and VcGluP in the absence of ligand (inset), (**C**) for BP0403 in the presence of several potential ligands, and (**D**) in the presence and absence of L-glutamate. Derivative curves shown are representative curves taken from a dataset of 4 technical repeats. Data from single experiments are shown, but each assay was performed on at least 2 occasions with the same overall outcome.

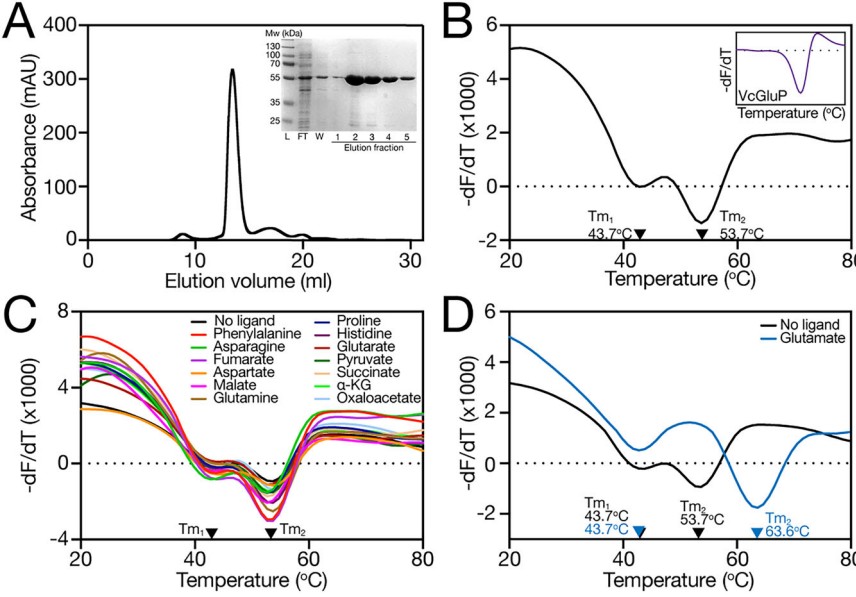

and BP0403$_{SBP}$ (residues 208–503). DSF analysis revealed that each isolated domain produces a single melt trough; BP0403$_{SBP}$ with a Tm of 61.6 °C, and BP0403$_{NTD}$ with a Tm of 39.8 °C (Fig. 3A). While neither of the Tms from the isolated domains corresponds exactly with the two Tms obtained from BP0403$_{FL}$, comparison of these data with the double dip DSF data of BP0403$_{FL}$ indicates that Tm$_1$ likely corresponds with the BP0403$_{NTD}$, and Tm$_2$ is likely the BP0403$_{SBP}$ (Fig. 3A).

To confirm that Tm$_2$ corresponds with the BP0403$_{SBP}$, we performed DSF individually on the BP0403$_{NTD}$ and BP0403$_{SBP}$ in the presence L-glutamate (Fig. 3B). While addition of 1 mM L-glu had essentially no effect on the Tm of BP0403$_{NTD}$, the Tm of the BP0403$_{SBP}$ was increased by 4 °C (Fig. 3B), substantially less than the 9.9 °C shift seen for BP0403$_{FL}$'s Tm$_2$ (Fig. 2D). To investigate this further, we performed an L-glutamate titration with BP0403$_{SBP}$ using DSF, which revealed the dose-dependent Tm increase expected for ligand binding to an SBP (Fig. 3C). To support the DSF-based ligand identification, we wished to perform tryptophan fluorescence spectroscopy on BP0403$_{FL}$. However, while BP0403$_{FL}$ contains 2 tryptophans, they are both located in the NTD, which does not bind ligand according to our DSF data. Specifically monitoring tyrosine fluorescence with BP0403$_{FL}$ was not possible due to the tryptophan emission masking any dose dependent fluorescence changes. Therefore, we monitored tyrosine fluorescence changes with the isolated SBP domain (Fig. 3D). Addition of 5 μM L-glutamate induced an 11% quench in the protein fluorescence, which was saturable, as there was no further quenching upon addition of 100 μM L-glutamate (Fig. 3D). We performed an L-glutamate titration that revealed a K$_d$ of 180 ± 32 nM for L-glutamate (Fig. 3D, inset).

To investigate the amino-acid binding determinants of BP0403$_{SBP}$, we generated an L-glutamate bound model of BP0403$_{SBP}$ using AlphaFold 3 and superimposed it with the X-ray structure of VcGluP, a TAXI SBP of conventional size from *Vibrio cholerae* that we previously showed stereoselectively binds L-glutamate with high-affinity and L-glutamine and pyroglutamate with lower affinity (Fig. 3E, Supplementary Fig. 4A–C for model quality evaluation)[22]. This structural comparison revealed an almost identical arrangement of several amino acids in the predicted substrate binding site (Fig. 3E), including Y219 and Q263, which are equivalent to the two most influential binding site residues in VcGluP (Y45 and Q93 in VcGluP)[22]. Based on these results and our previous characterisation of VcGluP, we predicted that Y219 and Q263 are key to glutamate binding in BP0403. To test this prediction, we individually mutated each residue to alanine and determined the glutamate binding affinity using tyrosine fluorescence. The resulting binding curves revealed that glutamate binding

was substantially reduced for both mutants; BP0403-Y219A exhibited a K$_d$ of 31 ± 6 μM, and BP0403-Q263A had a K$_d$ of 45 ± 3 μM (Supplementary Fig. 3). Compared to wildtype protein, mutating Y219 and Q263 to alanine resulted in a 167- and 250-fold reduction in binding affinity of wildtype, respectively, confirming the binding site position in BP0403.

### BP0403 is a dimer

Comparison of our DSF analysis of BP0403$_{FL}$ and BP0403$_{SBP}$ revealed that 1) the isolated SBP domain has a substantially higher Tm (~8 °C) than when it is part of the full-length protein, and 2) the ligand-induced ΔTm is much smaller for the isolated SBP domain than when part of BP0403$_{FL}$ (4 °C vs 9.9 °C, respectively). While it is not clear what is causing these differences in stability and ΔTm, it is possible that each domain in the full-length protein influences the stability of the other. It is also possible that BP0403$_{FL}$ is oligomeric and one melt trough results from dissociation of the oligomer and the second trough is caused be unfolding of the globular domains. Indeed, oligomerisation is a very rare, but not unheard-of arrangement for TRAP SBPs[31].

To investigate the oligomeric state of BP0403, we applied purified BP0403$_{FL}$ to a SEC column and compared its elution volume to the elution volumes of protein standards of known molecular weight (Fig. 4A). Analysis of the molecular weight of BP0403 using a standard curve generated from the molecular weight standards revealed that the elution volume corresponds to a 114.8 kDa protein (Fig. 4B). As the estimated molecular weight of BP0403 is 53.7 kDa based on the amino acid sequence, these data are consistent with BP0403 forming a stable dimer in solution.

Intrigued by this finding, we modelled the BP0403 dimer using AlphaFold 3[26]. Three different dimer arrangements were observed in the 479 best models (with a ranking score of 0.6 or above, Supplementary Fig. 5A). However, almost all of these models (96.4%) converged on a single dimeric arrangement, which also coincided with the arrangement depicted in the highest-scoring model, indicating AlphaFold 3 robustly and consistently predicted one dimeric arrangement. The highest scoring model for the dimer depicts a "domain-swapped" arrangement in which the NTD from one protomer appears to interact with the SBP domain of the second protomer; the second BP0403 protomer is arranged as a mirror image of the first (Fig. 4C, Supplementary Fig. 5B–D for model quality evaluation). While there are minimal contacts predicted between the 2 SBPs, a remarkable feature of this model is the arrangement of the helical domain, which form interlocking hairpins (Fig. 4C). The four helices contributing to this interface are amphipathic and most of the interfacial interactions are

**Fig. 3 | Ligand binding analysis on isolated SBP and NTD domains. A** Derivatives of the unfolding curves (dF/dT) for BP0403$_{FL}$ (black data), BP0403$_{SBP}$ (blue data) and BP0403$_{NTD}$ (orange data). **B** Derivatives of the unfolding curves (dF/dT) for BP0403$_{SBP}$ and BP0403$_{NTD}$ with (solid lines) and without (dashed lines) 1 mM L-glutamate. **C** Derivatives of the unfolding curves (dF/dT) for BP0403$_{SBP}$ with increasing concentrations of L-glutamate. **D** Tyrosine fluorescence scan of BP0403$_{SBP}$ in the absence of ligand (black data), 5 μM L-glutamate (pink data), 100 μM L-glutamate (cyan data). *Inset*: representative L-glutamate binding curve. Average Kd is shown ($n = 10$) and error represents SEM (all underlying binding curves are shown in Supplementary Fig. 3). **E** Super-imposition of the binding site residues of BP0403 modelled with bound L-glutamate using AlphaFold 3 (blue residues and yellow glutamate ligand) and the VcGluP crystal structure (light grey residues and orange glutamate ligand). pTM, ipTM, combined pTM_ipTM, Ranking score, LDDT representation and PAE matrix of best score model is in Supplementary Fig. 4A–C.

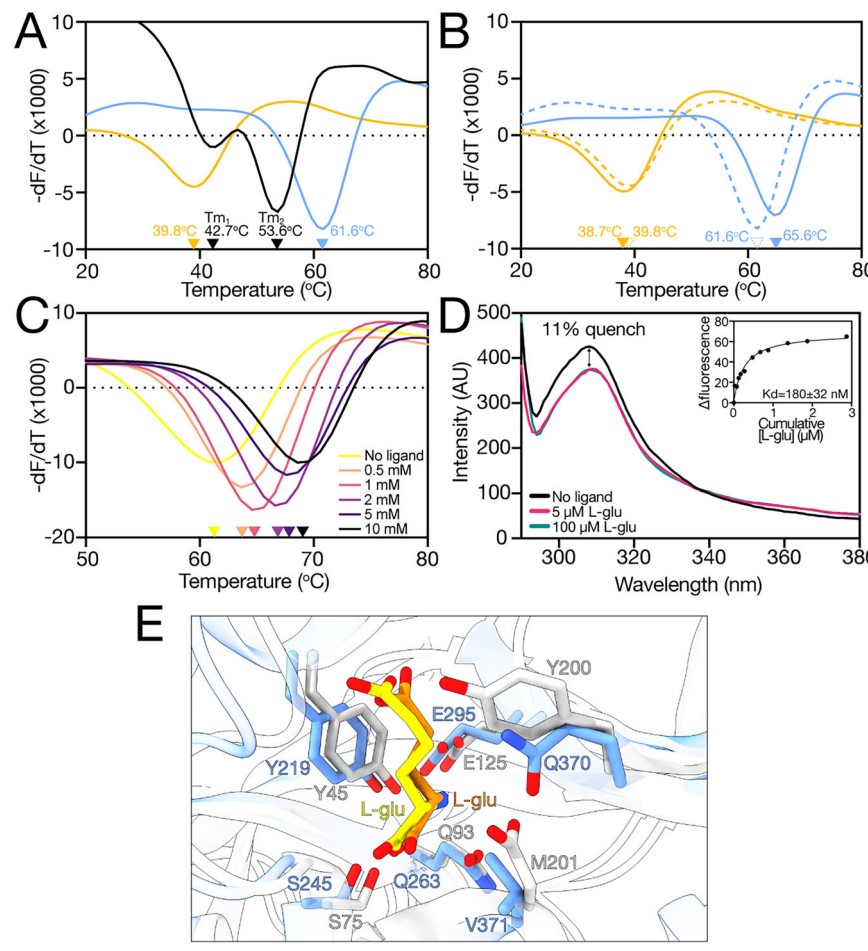

hydrophobic, resulting in a substantial interaction interface burying 1683 Å$^2$ surface area (Fig. 4D). Based on this model, we predict that the helical domain interaction is chiefly responsible for maintaining the dimeric arrangement of BP0403. To test this prediction, we sought to introduce mutations into the helical domain that would disrupt the interface packing. If the helical domain is indeed primarily responsible for the dimer formation, then such disruptive mutations would result in monomerisation of BP0403. To disrupt the interface, we replaced one of the interfacial leucine residues (L201) with an arginine (which would lead to two arginines at the interface of the homodimer, Fig. 5A), reasoning that the charged and relatively bulky character of an arginine in this position would be sufficient to destabilise the interface. We expressed and purified BP0403-L201R and SEC analysis revealed that, while maintaining a symmetrical peak indicative of stable protein, the L201R mutant eluted considerably later than the wildtype protein indicating a smaller molecular weight (Fig. 5B). Comparison with molecular weight standards revealed that L201R has a molecular weight of 64.4 kDa, indicating monomerisation of BP0403 (Fig. 5C). DSF analysis of monomerised BP0403-L201R revealed the same two trough melt curve exhibited by wildtype protein (Supplementary Fig. 5E, F). These data demonstrate that the monomerised protein is properly folded and that the characteristic double dip melt curve of BP0403 is not due to initial destabilisation of the dimer and then unfolding of each monomer. Rather, these data strongly support our hypothesis that the two troughs represent the separate unfolding of the NTD and SBP domains. Overall, these data strongly suggest that the helical domain is solely responsible for dimerisation of BP0403 and any interactions between the SBPs or between the NTDs and SBPs are inconsequential to oligomerisation.

Based on these findings, we would also predict that the isolated SBP and NTD domains are monomeric. To test this prediction, we generated

truncations of BP0403 and independently expressed either BP0403$_{SBP}$ or BP0403$_{NTD}$. We purified each of the independent domains and SDS-PAGE analysis revealed that each domain migrated in-line with its estimated molecular weight based on the amino acid sequences; 33.8 kDa and 16.4 kDa for BP0403$_{SBP}$ and BP0403$_{NTD}$, respectively (Fig. 5D, E). We further purified each domain using SEC and compared the elution volumes to those of the molecular weight standards used previously (Fig. 5F, G). This analysis revealed single peaks for BP0403$_{SBP}$ and BP0403$_{NTD}$ that correspond with proteins of 34.5 kDa and 18.9 kDa, respectively, demonstrating that the isolated domains are monomeric in solution and supportive of our model.

## BP0403 is not associated with a TRAP membrane component but with GltS

TRAP SBPs must interact with an integral membrane transporter to move substrates across the cytoplasmic membrane. While some TRAP SBP genes are 'orphans'[2], the vast majority are co-localised in an operon with the gene(s) expressing the TRAP membrane component(s). Therefore, to identify the cognate membrane component(s) to accompany BP0403, we analysed the genome context of *BP0403*. To our surprise, there were no genes encoding TRAP membrane proteins in the *BP0403*-containing operon or immediate gene region. Instead, *BP0403* is downstream of *BP0402*, which encodes *gltS*, with the end of *gltS* overlapping with the start of *BP0403* by 3 bp, indicative of translational coupling (Fig. 6A). GltS belongs to the glutamate:Na$^+$ symporter (ESS) family (TCDB ID: 2.A.27[32]), members of which are widespread in prokaryotes, but are not found in eukaryotes. Further analysis of genome context data using SeedViewer revealed a gene encoding GltS was also adjacent to genes encoding the elongated TAXI SBPs from multiple organisms (Fig. 6A)[33]. Furthermore, extending our

**Fig. 4 | Oligomeric state analysis of BP0403 and modelling of dimeric arrangement. A** SEC trace of BP0403 (red data) compared to 5 SEC standards of known molecular weight (grey data); thyroglobulin (T, 660 kDa), gamma-globulin (G, 150 kDa), ovalbumin (O, 43 kDa), ribonuclease A (R, 14 kDa) and p-aminobenzoic acid (P, 0.14 kDa). **B** Elution volume as a function of log MW of the SEC MW standards (grey) and BP0403 (red). **C** Cartoon representation of the highest scoring AlphaFold 3 model of the BP0403 dimer revealing a domain-swapped arrangement. The signal peptide has been omitted and lipid anchor site indicated. pTM, ipTM, combined pTM_ipTM, Ranking score, pLDDT representation and PAE matrix of best score model are in Supplementary Fig. 5B–D. Domains are coloured as in Fig. 1; protomer 1 is in lighter colours compared to protomer 2. **D** Close-up image of the helical domain interactions in domain-swapped model. Hydrophobic residues are highlighted in yellow.

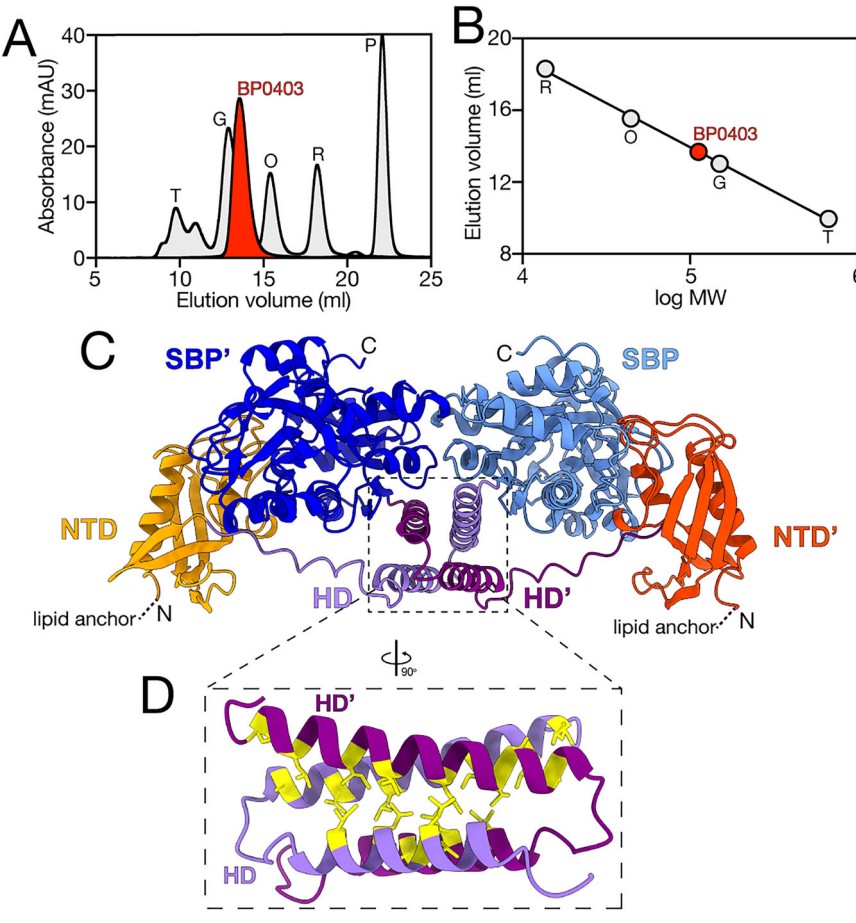

search to include TAXI SBPs of conventional size, we found several operons in which a TAXI SBP gene was directly up- or downstream of a gene encoding GltS (Fig. 6B). Importantly, this genetic association is found in multiple disparate organisms across the prokaryotic phylogeny, strongly suggesting that the TAXI SBP and GltS are functionally coupled. Low resolution cryo-EM and blue native PAGE of *E. coli* GltS revealed that it is a dimer and functional characterisation showed preferential transport of L-glutamate, with D-glutamate and 2-methyl glutamate transported at much lower affinity[34,35], strengthening the link with our dimeric L-glutamate-specific TAXI SBP, BP0403.

As there are no GltS structures available, we modelled the GltS dimer using AlphaFold 3 (Supplementary Fig. 6A–C for model quality evaluation)[26], which revealed the presence of features strongly reminiscent of the Na$^+$:succinate transporter VcINDY and TRAP transporter membrane components; namely, a scaffold domain and a transport domain characteristic of an elevator type mechanism (Fig. 6C)[16,17]. In addition, GltS has long been known to share a similar topology with the Na$^+$:citrate transporter CitS, a member of the 2-hydroxycarboxylate (2HCT) family[36], which operates via an elevator mechanism[37,38]. Based on the observation that both BP0403 and GltS are dimeric, we hypothesise that, upon binding glutamate, BP0403 presents the substrate to GltS, which uses an elevator mechanism similar to TRAP transporters, VcINDY and CitS to transport the substrate across the membrane.

To investigate the possible structural arrangement of the heterotetrameric BP0403:GltS complex, we modelled it using AlphaFold 3 (Supplementary Fig. 6D–F for model quality evaluation)[26]. In this model, the BP0403 dimer remains identical to the BP0403 dimer modelled alone (Fig. 6D). For each BP0403 protomer in the tetrameric complex, the NTD interacts with the transport domain of one GltS protomer and the SBP domain interacts with the other GltS protomer. The helical domains overlay

the scaffold domain where they intertwine as in the BP0403 dimer model. In the physiological setting, BP0403 would be attached to a lipid headgroup via an N-terminal cysteine residue, and our tetrameric model positions this cysteine residues precisely where we predict the lipid headgroups would arrange, supporting the feasibility of such an arrangement (Fig. 6D). Both SBP domains in our model are positioned above the scaffold:transport domain interface, which is the likely position of the GltS binding site. Therefore, our tetrameric BP0403:GltS model provides a mechanically and structurally feasible way for these proteins to work in concert during transport. Interestingly, while the GltS dimer modelled alone depicts an inward-facing state conformation in which the binding site is positioned on the cytoplasmic side of the membrane, the GltS dimer in the heterotetramer is modelled in the outward-facing state (Fig. 6D). As predicted based on similar structural features to other known elevator-like transporters, comparison of the two GltS models reveals rigid body movement of the transport domain in an elevator-like movement (Supplementary Fig. 7).

While the functional coupling between the TAXI SBPs and GltS needs to be experimentally verified, our hypothesis is substantially strengthened by the existence of several examples of *TAXI-gltS* gene fusions in genomes of the bacterial family Lachnospiraceae, which result in an N-terminal TAXI SBP domain fused to a C-terminal GltS (Fig. 6E). Presumably in this situation, GltS would still dimerise leading to a system reminiscent of the one we propose for BP0402/03.

## Discussion

In this work, we have described the discovery and characterisation of a novel type of glutamate-binding SBP in the TAXI family with a domain architecture completely distinct from all other TAXI proteins characterised to date. Moreover, it is genetically linked with, and translationally coupled to, the Na$^+$:glutamate symporter GltS and not with a conventional TRAP

**Fig. 5 | Monomerising BP0403 and oligomeric state analysis of isolated SBP and NTD domains.**
**A** Cartoon representative of helical domain with position L201 highlighted in both protomer (yellow residue). **B** SEC traces of BP0403-L201R (yellow data) compared to BP0403 wildtype (red data) and 5 SEC standards of known molecular weight (grey data). **C** Elution volume as a function of log MW of the SEC MW standards (grey data), BP0403-L201R (yellow data), and BP0403 wildtype (red data). **D**, **E** SDS-PAGE analysis of the IMAC purification fractions for BP0403$_{NTD}$ and BP0403$_{SBP}$, respectively. Lane labels are the same as Fig. 2A. **F** SEC trace of BP0403$_{SBP}$ (blue data) and BP0403$_{NTD}$ (orange data) compared to 5 SEC standards (grey data). **G** Elution volume as a function of log MW of the SEC MW standards (grey data), BP0403$_{SBP}$ (blue data), and BP0403$_{NTD}$ (orange data). Molecular weight standards are the same as used in Fig. 4.

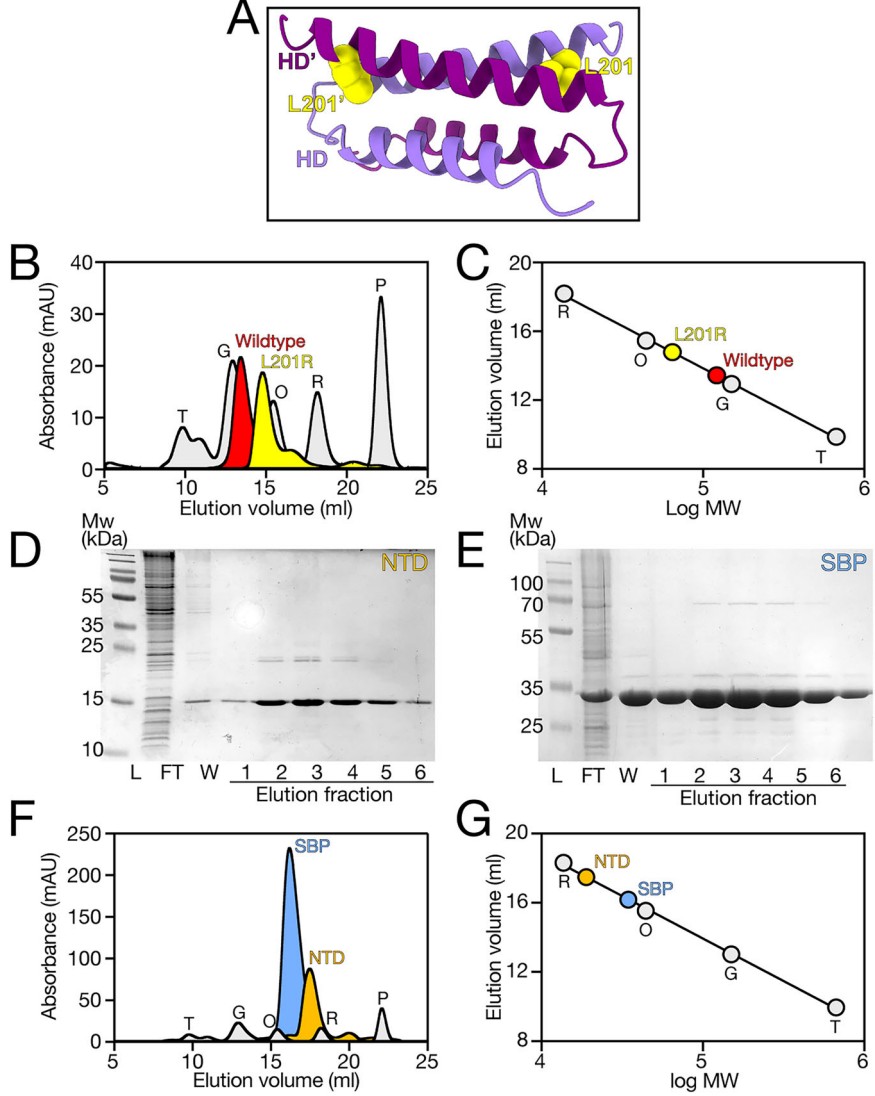

transporter. Our results, combined with the observation that some bacteria possess single protein TAXI-GltS fusions, strongly suggest that this unprecedented arrangement represents an entirely novel type of binding-protein dependent secondary transporter.

BP0403 is much larger than conventional TAXI SBPs. We have shown by AlphaFold modelling and biochemical analysis that it is composed of separate and independently folded domains at the N- and C-termini, linked by a central helical domain. The distinctive double trough in the DSF profile of the full-length protein was suggestive of independent melting events of two domains with different Tm values, which we confirmed by analysis of the isolated N- and C-terminal domains and a monomerized mutant. Evidence from both structural alignment of the C-terminal domain with the conventional-sized TAXI SBP VcGluP and its ability to bind substrate, unequivocally identifies this domain as the SBP element of the protein. Our modelling indicates that the helical domain forms the dimerization interface, which is supported by molecular weight analysis of the SBP and NTD in isolation and our ability to monomerize full-length BP0403 with a single amino acid substitution in the helical domain. An outstanding question is what function the NTD serves. Analysis using Foldseek revealed an overall similar fold to the core domain of PilO, which is a component of the Type IV pilus machine (Supplementary Fig. 8)[39]. PilO plays a structural role in connecting different parts of the pilus machinery and forms multiple protein:protein interactions[40,41]. In our model of the BP0403:GltS

heterotetramer (Fig. 6D), the NTD is predicted to make a substantial number of contacts with the dynamic transport domain of GltS as well as other regions of BP0403, suggesting a structural bridging role in this instance too. However, it remains unclear why this structural bridge is required for the BP0403:GltS interaction when it is absent from all previously characterised SBP-dependent transporters.

In contrast to DctP-type SBPs, the substrates bound by TAXI proteins are far less well characterised. The first TAXI SBP structure to be determined had a bound ligand consistent with either glutamate or glutamine[21], and subsequent characterisation of other TAXI SBPs demonstrate or suggest specificity for L-glutamate[22,42], indicating this dicarboxylic amino-acid is a common ligand in this family. This is in keeping with our identification by DSF of L-glutamate as the ligand for BP0403. Moreover, we observed conservation of the key Y and Q residues in BP0403 that were shown to be crucial for L-glutamate binding in VcGluP[22]. However, characterisation of a TAXI-TRAP system from *Proteus mirabilis* revealed specificity for both 2-oxoglutarate and 2-hydroxyglutarate[23], and a system from *Azoarcus* transports orthophthalic acid[43], hinting at a greater diversity of substrates.

Most GltS proteins operate as Na$^+$:glutamate symporters without the need for interaction with an SBP. BP0402 is the sole member of the ESS family present in *B. pertussis* but we do not yet know if it can operate independently of BP0403. What could be the physiological advantage of the

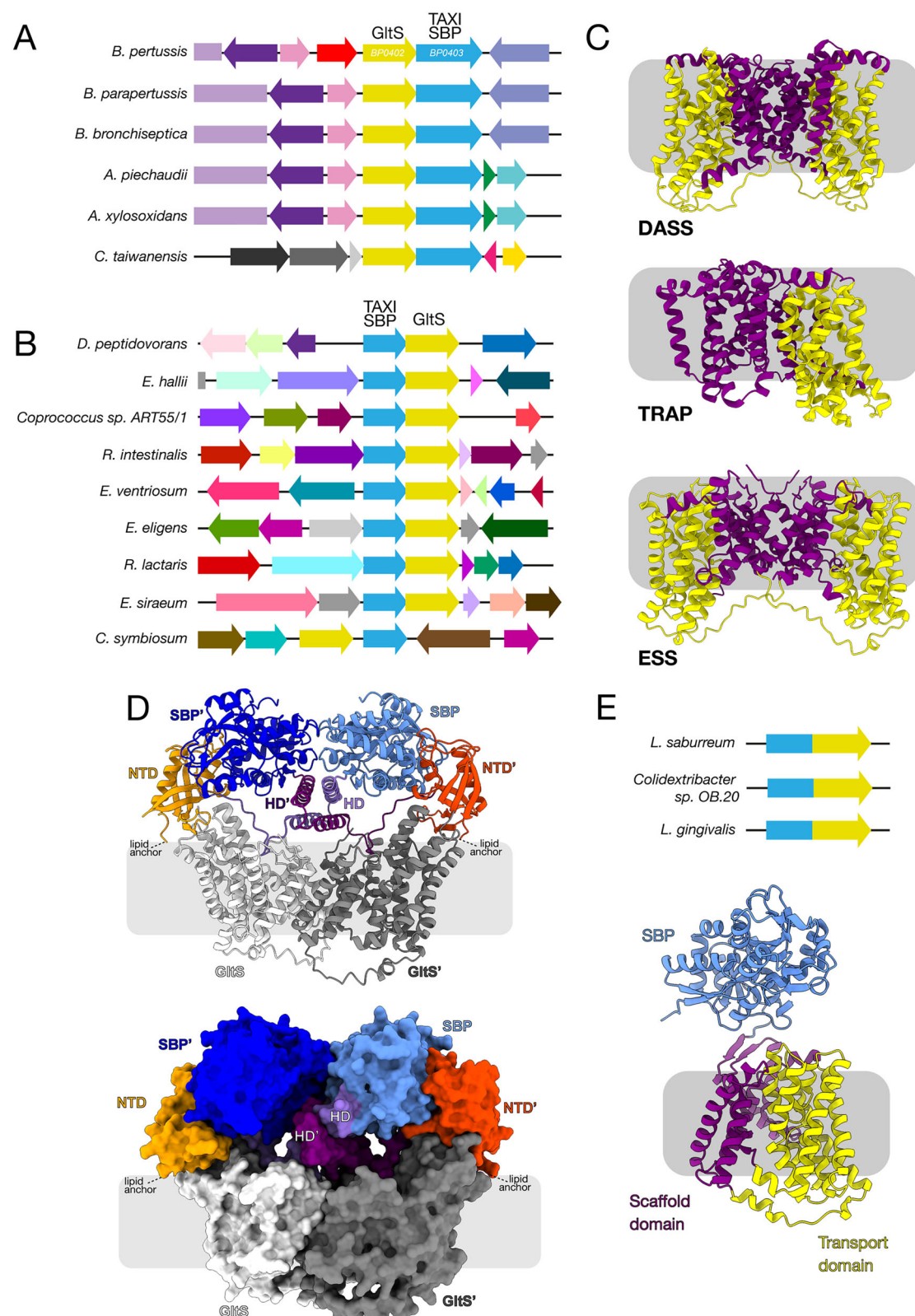

interaction of these proteins? One possibility is that the SBP could bind a different ligand, thus effectively changing the substrate specificity of the membrane transporter. However, as we have shown that BP0403 binds L-glutamate and all characterised GltS systems transport glutamate, we find this unconvincing. For scarce nutrients, SBPs confer a clear advantage by increasing the encounter rate between substrate and transporter, which can lead to enhanced rates of transport compared to the absence of an SBP[44]. In addition, recruitment of the TAXI protein will confer a higher affinity to the transporter compared with GltS alone. Indeed, our tyrosine fluorescence titrations of the isolated SBP indicate a sub-μM Kd for L-glutamate, whereas the Km of *E. coli* GltS for L-glutamate in membrane vesicles is 13.5-30 μM[35,45].

**Fig. 6 | Linkage between genes encoding TAXI SBPs and GltS, and modelling of the TAXI-GltS interaction. A** Genome context of BP0403 and homologues (blue genes) from 5 other bacteria showing close proximity of gene encoding GltS (yellow gene). **B** Genome context of genes encoding TAXI SBPs of conventional size (blue gene) and genes encoding GltS (yellow gene). **C** Structural comparison of DASS transporter crystal structure (VcINDY, PDB: 5UL7), TRAP transporter cryo-EM structure (HiSiaQM, PDB: 7QE5), and ESS AlphaFold 3 model (BpGltS). Coloured to highlight scaffold (purple) and transport (yellow) domains. pTM, ipTM, combined pTM_ipTM, Ranking score, pLDDT representation and PAE matrix of the best scoring GltS dimer model is in Supplementary Fig. 6A–C. **D** Cartoon representation (top) and surface representation (bottom) of the heterotetrameric BP0403:GltS complex modelled using AlphaFold 3. The domains of BP0403 are colour coded and labelled as in Fig. 4. The GltS protomers are coloured white and dark grey. pTM, ipTM, combined pTM_ipTM, Ranking score, pLDDT representation and PAE matrix of best score model is in Supplementary Fig. 6D–F. **E** Examples of fusion between TAXI and *gltS* genes in the family Lachnospiraceae. Bottom: AlphaFold2 structural model of *Lachnoanerobaculum saburreum* fused TAXI-GltS. Coloured to highlight scaffold (purple), transport domain (yellow) and SBP (blue).

A higher affinity transporter would be advantageous in the scavenging of scarce nutrients. In the tissues of mammalian hosts, extracellular L-glutamate concentrations are relatively low as this amino-acid is rapidly transported and metabolised by eukaryotic cells[46]. As *B. pertussis* adheres to and grows on cell surfaces, the association of GltS with a TAXI protein to scavenge glutamate could result in a competitive advantage for this pathogen. *B. pertussis* cannot utilise carbohydrates as a carbon source, primarily using a small selection of amino-acids, of which glutamate has the greatest enhancements in growth rate and cell yield[47–49]. Glutamate availability modulates gene expression in *B. pertussis*[50], with glutamate limitation up-regulating key virulence factors, such as the type 3 secretion system (T3SS), as well as enhancing adhesion[46]. In keeping with its relative importance, *B. pertussis* possesses multiple potential routes for glutamate uptake, including the TAXI-GltS system identified here and at least one ABC system with an SBP (BP3831)[51]. The structures of two DctP-type SBPs (BP1887 and BP1891) of uncharacterised TRAP transporters have been determined, both of which bind pyroglutamate[52], which can be converted to glutamate by pyroglutamase[53]. Finally, *B. pertussis* encodes a large number of genes encoding SBPs of the tripartite tricarboxylate transporter (TTT) family[54], one of which (BugE) binds glutamate, implying the presence of a TTT-type glutamate uptake system[55]. Interestingly, all 3 proposed transport routes employ SBPs, consistent with the need to scavenge this nutrient.

In a diverse range of bacteria, we found genes for both elongated and conventional sized TAXI SBPs closely linked to *gltS* (Fig. 6A and B). However, in some bacteria we identified TAXI-GltS gene fusions (Fig. 6E). Examination of the INTERPRO database reveals that of ~10,000 entries for GltS, only 14 have this TAXI-GltS fusion architecture and the majority of these are found in members of one family, the Lachnospiraceae. These bacteria are poorly characterised members of the gut microbiome but they are firmicutes which therefore (presumably) do not have a conventional periplasm. Thus, the evolution of a TAXI-GltS fusion protein may be rationalised as one solution to ensure tethering of the SBP to the membrane transporter in these bacteria.

Prior to this study, the only families of secondary transporters known to operate with a soluble SBP were the TRAP and TT transporters. Why are SBPs not more widely associated with other diverse types of secondary transporters? It is now known that TRAP transporters operate by an elevator type mechanism. TT transporters are structurally very similar and likely operate in an analogous way. Our modelling suggests that GltS has clearly discernible scaffold and transport domains, implying that it too is likely to work as an elevator. This might suggest that some intrinsic feature(s) of the elevator mechanism is required to allow an associated SBP to bind and release its ligand as part of the transport cycle. The identification of further novel classes of SBP dependent secondary transporters will allow this proposal to be tested.

## Data availability
The source data for all graphical data is in Supplementary Data 1. All other data are available from the corresponding author on reasonable request.

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

## Acknowledgements

D.J.K. acknowledges the receipt of a Leverhulme Emeritus Fellowship (EM/2024-005) from the Leverhulme Trust, UK. This work was financially supported by a Biotechnology and Biological Sciences Research Council (BBSRC) grant (BB/V007424/1) awarded to CM and the SoCoBio BBSRC doctoral training partnership (BB/T008768/1). V.L. acknowledges that this research was completed in part with computational resources and technical support provided by the Research Computing Center at the Medical College of Wisconsin.

## Author contributions

L.M.J., J.J.S.T. purified proteins and collected binding assay and oligomeric state data. J.F.S.D. performed additional binding assays. C.M., J.F.S.D. and D.J.K. performed the bioinformatic analysis. V.L. modelled structural complexes and evaluated model quality and C.M. analysed the structural models. C.M. supervised the research. C.M., V.L. and D.J.K. wrote the manuscript. All authors read and edited the manuscript.

## Competing interests

The authors declare no competing interests.
