## [Transparent Peer Review file · Communications Biology]

A new class of binding protein-dependent solute transporter exemplified by the TAXI-GltS system from *Bordetella pertussis*

Corresponding Author: Dr Christopher Mulligan

Version 0:

Reviewer comments:

Reviewer #2

(Remarks to the Author)

In their manuscript, Jaques et al. characterized an unusual substrate binding protein (SBP) from the TAXI TRAP family that is thought to interact with a transporter from the GltS family of Na⁺-coupled glutamate symporters.

By analyzing TAXI proteins using sequence alignments, the team observed that the SBP, BP0403, is substantially larger than other TAXI SBPs. They recombinantly expressed and successfully purified the SBP and used this protein to identify the substrate by elegant differential scanning fluorometric screening experiments. Structural predictions and the gel filtration chromatogram suggested that the wild-type protein forms dimers, which could be validated by biochemical analyses of different SBP constructs.

Overall, I like the manuscript and the biochemical data are solid. I think the AF predictions need to be discussed a bit more carefully and their limitations need to be made clearer.

Major:

1) In Figure 3E, you have highlighted the amino acid residues postulated to form the glutamate binding site. This is based on alignment with another glutamate SBP and is reasonable, but could you verify the prediction by mutating these key residues and observing a loss of binding? Also, the figure is not very clear. I would suggest using different colors for different types of atoms. The glutamate is not labeled and is shown in an orientation that makes it a bit difficult to see the position of the side chain and main chain atoms. The side chain carboxylate does not seem to interact with anything. Is this really the case? The structural overlay does not seem to be very good, is this because the AF model is in the open state? Is this the reason for the rather high r.m.s.d. (line 172)? Maybe you could split the AF model into rigid bodies (e.g. its two lobes) and align them with VcGluP to get a better alignment. Alternatively, you could also predict the complex structure with Chai-1 (a server version is available, you can add the glutamate as a smiles string). This is also possible with a local installation of AF3, but it is not trivial to make it work.

2) In lines 296 - 298, the authors argue that the differences in the T_m values of the individual domains compared to the full-length protein are most likely caused by a "fundamental difference in the arrangement of the SBP domain when isolated". What exactly do they mean by arrangement? A different conformation? The changes could also be caused by a stabilizing effect that one domain has on the other, or that one trough forms when the dimer dissociates and the other when the monomers unfold. Please add these possibilities to the text.

3) Figure 4 shows the AF model of dimeric SBP with magnification of the dimerization interface formed by the helical domain. Please show the pAE interaction matrix and the pTM and ipTM scores in the figure. I would also suggest showing the pLDDT scores in C. The authors describe how they used a truncated protein construct to investigate this prediction (lines 317-320). Why don't you use the full-length protein (minus the membrane anchor) and mutate residues in the predicted interface (e.g. charge flips, bulky amino acids) to directly support the model.

4) In line 521 - 547 you discuss the importance of glutamate transport for *B. pertussis* and speculate to underline the hypothesis of an advantageous high affinity transport. This section is very elaborate and I would suggest to shorten it.

- 5) Figure 6: I would suggest showing the pLDDT scores and the pAE interaction matrix in these figures. I tried to predict the complex with AF3, and while the overall architecture is predicted quite similarly, the quality scores are not very high.
- 6) Discussion: Line 490-492. Did you try the foldseek server? I found some structural similarities even to characterized proteins.
- 7) Discussion: Line 506. This paper may be of interest: Bosdriesz, E., Magnúsdóttir, S., Bruggeman, F. J., Teusink, B. & Molenaar, D. Binding proteins enhance specific uptake rate by increasing the substrate-transporter encounter rate. *FEBS J.* 282, 2394-2407 (2015).

Minor:

- 1) Line 14: There is a typo, it should be "Tripartite ATP-independent periplasmic transporter" ;-)
- 2) Line 204: shouldn't it be "due to increased protein stability"?
- 3) Figure 2: I would appreciate to find the annotations "Tm1 and Tm2" from panel D also in panel C
- 6) Line 218/219: could you please specify if the mean values from the two independent experiments are shown, or whether only one experiment is shown, but a second looks pretty much the same?
- 7) Line 377: please include the reference for "SeedViewer"
- 8) Line 460: "C" should be bold.
- 9) Methods: Please give the UNIPROT ids in the molecular biology section, too.

Reviewer #3

(Remarks to the Author)

This research from Mulligan and colleagues explores a novel, unusually large, and dimeric TAXI SBP from *Bordetella pertussis* (BP0403), diverging from traditional TRAP transporter genes. This protein binds glutamate but associates with a nontraditional TRAP membrane component, a member of the unrelated GltS family of Na⁺-dependent symporter (GltS). This suggests the identification of a previously uncharacterized SBP-dependent secondary transporter type. The novel arrangement of the TRAP transporter, supported by experimental testing, genome context analysis, and protein modeling, could provide valuable insights into bacterial nutrient uptake mechanisms and pathogenesis. The manuscript is generally well-designed and written; however, several critical issues need to be addressed.

Major comments

The authors initially used thermal stability as a preliminary method to characterize glutamate binding of the full-length BP0403, a dimeric tri-domain protein that includes the SBP domain. From the titration experiment, one can estimate a K_d value of around 1.5 mM, (extrapolated from Fig. 3C). They then employed a more precise method, intrinsic fluorescence, to assess ligand binding. However, due to technical limitations, they could only apply this technique to the isolated SBP domain, which appears to be a monomer. This approach yielded a much higher binding affinity with a K_d of 0.18 μM. While the thermal stability method may be less accurate because temperature increases can skew results, leading to higher K_d values, the significant 10,000-fold difference between the two techniques is difficult to reconcile. One may argue that the native full-length dimeric tri-domain protein – the form responsible for the biological function – is much less effective in binding glutamate than the isolated monomeric SBP domain.

What is the model equation used to determine the K_d? This is not reported. The authors should investigate the binding mechanism more thoroughly, especially considering the novelty of the quaternary structure of the protein. The binding could involve allosteric or sequential mechanisms, which would require more complex models to accurately describe the binding process. which could be allosteric or sequential. Additionally, to improve the clarity and reliability of the results, the authors should provide a fitting curve with error bars that reflect the variability of replicates. This would allow for a more accurate representation of the data and help assess the precision of the K_d determination. A representative curve without error bars can be misleading, especially when interpreting binding affinities from a single dataset.

The methods section could benefit from additional details to enhance reproducibility. For instance, the specific buffer used in the intrinsic fluorescence (ITF) assays should be clearly stated, as buffer conditions can significantly affect the outcome of binding studies. Including information such as buffer composition, pH, and ionic strength will allow others to replicate the experiments accurately.

Regarding the modeling tests with AlphaFold, more details on the parameter settings are essential. Specifically, the authors should mention:

The number of cycles used in the AlphaFold prediction process. This information is critical for understanding the level of convergence and whether additional cycles might improve accuracy.

Whether the highest scores (e.g., pLDDT or predicted local distance difference test scores) reached saturation. If the scores plateaued, this would indicate that further iterations wouldn't improve the model's accuracy. Conversely, if the scores did not saturate, it could suggest that additional cycles could refine the model further.

Minor comments

Glutamate, along with the other potential ligands, should also be included in Figure 2C.

Fig. 1A: choose a different color for the HD interdomain to better distinguish it from the adjacent SBP domain.

Line 14. The Abstract starts with Tripartite ATP-dependent periplasmic (TRAP) transporter...an oversight about the ATP-dependence statement must be corrected. (see the discrepancy with line 45).

Line 41. A serial comma after "high affinity" might improve readability.

Line 68. "relatively poorly characterized" is slightly redundant and confusing; I suggest killing the adverb "relatively."

Lines 71-74. The sentence encompassing lines 71-74 is long and could be split for readability.

Line 79. A comma after "level" is necessary.

Line 83. "While there are examples of orphan TRAP SBPS..." add reference(s) for this sentence.

Line 373. It seems that the overlap of the two genes is by 4 bp. Please re-check.

Line 377. A Reference must be provided for SeedViewer tool.

Line 540-542. To improve clarity and ensure the most up-to-date references are used, the outdated reference (Ref. 50, which is around 20 years old) should be replaced when introducing the Bordetella Uptake genes (Bug) of the tripartite tricarboxylate transporter (TTT) family in the discussion. Instead, use the more recent 2024 reference: DOI: 10.1002/2211-5463.13876. Since Ref. 50 is more relevant to the subsequent sentence; it should remain there where it is correctly cited. This ensures that the discussion is both current and appropriately referenced.

Version 1:

Reviewer comments:

Reviewer #2

(Remarks to the Author)

The authors have addressed all my concerns and I support publication.

Reviewer #3

(Remarks to the Author)

The authors have duly addressed my concerns. I have no further comment.

Response to reviewers' comments

We would like to thank both reviewers for the time they took to carefully evaluate our manuscript. We also thank the reviewers for their insightful suggestions. We believe these suggestions, and the additional work and edits they prompted, have greatly improved the manuscript.

We have addressed each of the reviewer's comments point-by-point below. Reviewer's points are in black, our responses are in blue.

Reviewer #2 (Remarks to the Author):

In their manuscript, Jaques et al. characterized an unusual substrate binding protein (SBP) from the TAXI TRAP family that is thought to interact with a transporter from the GltS family of Na⁺-coupled glutamate symporters.

By analyzing TAXI proteins using sequence alignments, the team observed that the SBP, BP0403, is substantially larger than other TAXI SBPs. They recombinantly expressed and successfully purified the SBP and used this protein to identify the substrate by elegant differential scanning fluorometric screening experiments. Structural predictions and the gel filtration chromatogram suggested that the wild-type protein forms dimers, which could be validated by biochemical analyses of different SBP constructs.

Overall, I like the manuscript and the biochemical data are solid. I think the AF predictions need to be discussed a bit more carefully and their limitations need to be made clearer.

Major:

1) In Figure 3E, you have highlighted the amino acid residues postulated to form the glutamate binding site. This is based on alignment with another glutamate SBP and is reasonable, but could you verify the prediction by mutating these key residues and observing a loss of binding?

We have now characterised glutamate binding affinity of BP0403-Y219A and BP0403-Q263A, both of which result in a substantial reduction in glutamate binding affinity; 167- and 250-fold for Y219A and Q263A, respectively. This is in line with the reduction in binding affinity seen for the equivalent mutations for VcGluP. This new analysis is included in the main text lines 314-322, and the underlying binding curves for the mutants are shown in SI fig 3.

Also, the figure is not very clear. I would suggest using different colors for different types of atoms.

The glutamate is not labeled and is shown in an orientation that makes it a bit difficult to see the position of the side chain and main chain atoms.

We have changed the colour scheme to add contrast between the 2 structures and coloured them by atom type. We have reoriented the view to better visualise the whole of the glutamate. We have also labelled glutamate. In addition, the new Figure 3E is now

a comparison between the binding sites of VcGluP and BP0403 SBP with glutamate modelled into the latter using AlphaFold 3 (described in more detail below).

The side chain carboxylate does not seem to interact with anything. Is this really the case?

No, this is not the case. As detailed in our previous study on VcGluP (Davies et al, JGP, 2024), the side chain carboxylate of the glutamate ligand makes main chain interactions with 2 residues and multiple hydrogen bonds with water molecules. As we cannot replicate these water:ligand interactions in our AlphaFold model of BP0403, we omitted these interactions for clarity.

The structural overlay does not seem to be very good, is this because the AF model is in the open state? Is this the reason for the rather high r.m.s.d. (line 172)? Maybe you could split the AF model into rigid bodies (e.g. its two lobes) and align them with VcGluP to get a better alignment. Alternatively, you could also predict the complex structure with Chai-1 (a server version is available, you can add the glutamate as a smiles string). This is also possible with a local installation of AF3, but it is not trivial to make it work.

We thank the reviewer for the suggestion. We have kept the original comparison of VcGluP and BP0403_{SBP} (line 190-192, Fig 1) because it was how we first identified that they have the same overall structure and that the C-terminal domain of BP0403 constitutes the SBP domain. However, based on the Reviewer's suggestion, we have now modelled BP0403 with bound L-glutamate using AF3 to better visualise the predicted binding site, which we use when discussing the binding site. We generated 250 models of BP0403 bound to glutamate with AF3 (see SI fig. 4A-C for quality assessments). We have replaced the superimposition in Fig 3E with a new alignment of VcGluP and liganded BP0403_{SBP}. As the reviewer will see from the new figure, the binding site is very similar to that of VcGluP as previously predicted. However, the inclusion of the ligand makes this much more satisfying, and we thank the reviewer for prompting us to include this extra analysis. We have added new text to line 306 to introduce this new alignment.

2) In lines 296 - 298, the authors argue that the differences in the T_m values of the individual domains compared to the full-length protein are most likely caused by a "fundamental difference in the arrangement of the SBP domain when isolated". What exactly do they mean by arrangement? A different conformation?

By "different arrangement", we were alluding to the possibility that it oligomerises, which we then explained in the following sentence (line 300-301 in original submission). However, we understand the Reviewer's point that this wording could be confusing. To clarify this section, we have deleted the offending sentences.

The changes could also be caused by a stabilizing effect that one domain has on the other, or that one trough forms when the dimer dissociates and the other when the monomers unfold. Please add these possibilities to the text.

We have now added these possibilities to the text (lines 327-331). However, we think we can now rule out the possibility that one trough is due to the dimer dissociating and the other is due to the monomers unfolding. Using the fully monomerised mutant we developed to address one of the Reviewer's points below, we performed DSF and

observed the same 2 troughs as wildtype. Therefore, we believe that the two melt events represent the NTD and SBP unfolding. We have included this new data (SI Fig. 5E, F) and a description of these findings in the main text (Lines 399-407).

3) Figure 4 shows the AF model of dimeric SBP with magnification of the dimerization interface formed by the helical domain. Please show the pAE interaction matrix and the pTM and ipTM scores in the figure.

In addressing this comment, we reprocessed the models for the BP0403 dimer and the BP0403/GltS heterotetramer using the AlphaFold 3, which was recently made available. Interestingly, the highest-scoring model in this new analysis with AF3 predicts a *different* arrangement of the BP0403 globular domains compared to our original prediction. As detailed in the revised manuscript, this new dimer model predicts the same extensive helical domain interaction, but has a domain swapped arrangement suggesting intermolecular interactions between the NTD of one protomer with the SBP of the other protomer (see Fig. 4). The model of the BP0403 dimer is now in the same arrangement as when modelled in complex with the GltS dimer. Importantly, this difference in predicted domain arrangement does not change any of our main conclusions.

The pTM, ipTM and ranking score values, the PAE matrix are now shown for each of the AF3 models. We have also included a model coloured by pLDDT for each model.

SI Fig. 4A-C relate to the glutamate-bound BP0403 monomer model.

SI Fig. 5A-D relate to the BP0403 dimer model.

SI Fig. 6A-C relate to the GltS dimer model.

SI Fig. 6D-F relate to the BP0403:GltS heterotetramer.

The quality assessment figures are signposted in the figure legends and in the main text itself.

I would also suggest showing the pLDDT scores in C.

We have added a figure of the best model coloured based on pLDDT score in SI Figures 4-6.

The authors describe how they used a truncated protein construct to investigate this prediction (lines 317-320). Why don't you use the full-length protein (minus the membrane anchor) and mutate residues in the predicted interface (e.g. charge flips, bulky amino acids) to directly support the model.

We thank the reviewer for this suggestion, which was something that we already had underway. We have included new data in Figure 5 showing that substituting L201 for an arginine disrupts the helical domain interface sufficiently to prevent oligomerisation. We believe that, combined with the monomeric nature of the independent domains, these data strongly support our conclusion that the dimer interface is coordinated solely by the helical domain. We have included the new data in Figure 5A-C and new text describing this experiment in the main text (lines 387-399).

4) In line 521 - 547 you discuss the importance of glutamate transport for *B. pertussis*

and speculate to underline the hypothesis of an advantageous high affinity transport. This section is very elaborate and I would suggest to shorten it.

We have now shortened this section of the Discussion (line 611-630).

5) Figure 6: I would suggest showing the pLDDT scores and the pAE interaction matrix in these figures. I tried to predict the complex with AF3, and while the overall architecture is predicted quite similarly, the quality scores are not very high.

Indeed, obtaining higher-quality models of the tetramer required extensive sampling. Of the total 1500 models we have 44 and 108 structural predictions above a 0.7 combined pTM_ipTM score and ranking score, respectively (see SI Fig. 6D). We have added in the same figure a representation of the best model, coloured by pLDDT score and the PAE matrix (SI Fig 6E,F).

6) Discussion: Line 490-492. Did you try the foldseek server? I found some structural similarities even to characterized proteins.

We had not tried Foldseek in the original submission. We have now performed this analysis and identified structural similarity to PilO, which is a structural element in bacterial pilus machinery. We have included a new supplementary figure (SI Fig. 8) showing a superimposition of the BP0403_{NTD} and PilO, and text describing the analysis in the Discussion (lines 573-582). We thank the reviewer for the suggestion.

7) Discussion: Line 506. This paper may be of interest: Bosdriesz, E., Magnúsdóttir, S., Bruggeman, F. J., Teusink, B. & Molenaar, D. Binding proteins enhance specific uptake rate by increasing the substrate-transporter encounter rate. FEBS J. 282, 2394-2407 (2015).

Thank you very much! We have now incorporated this into our discussion (line 601).

Minor:

1) Line 14: There is a typo, It should be “Tripartite ATP-independent periplasmic transporter” ;-)

Thank you for catching this one! It has now been corrected.

2) Line 204: shouldn't it be “due to increased protein stability”?

Changed to “stability”.

3) Figure 2: I would appreciate to find the annotations “Tm1 and Tm2” from panel D also in panel C

We have added “Tm1” and “Tm2” to panel C.

6) Line 218/ 219: could you please specify if the mean values from the two independent experiments are shown, or whether only one experiment is shown, but a second looks pretty much the same?

We have now specified this in the text.

7) Line 377: please include the reference for “SeedViewer”

Thank you for pointing this out. Reference added.

8) Line 460: “C)” should be bold.

Fixed.

9) Methods: Please give the UNIPROT ids in the molecular biology section, too.

UNIPROT IDs added.

Reviewer #3 (Remarks to the Author):

This research from Mulligan and colleagues explores a novel, unusually large, and dimeric TAXI SBP from *Bordetella pertussis* (BP0403), diverging from traditional TRAP transporter genes. This protein binds glutamate but associates with a nontraditional TRAP membrane component, a member of the unrelated GltS family of Na⁺-dependent symporter (GltS). This suggests the identification of a previously uncharacterized SBP-dependent secondary transporter type. The novel arrangement of the TRAP transporter, supported by experimental testing, genome context analysis, and protein modeling, could provide valuable insights into bacterial nutrient uptake mechanisms and pathogenesis.

The manuscript is generally well-designed and written; however, several critical issues need to be addressed.

Major comments

The authors initially used thermal stability as a preliminary method to characterize glutamate binding of the full-length BP0403, a dimeric tri-domain protein that includes the SBP domain. From the titration experiment, one can estimate a K_d value of around 1.5 mM, (extrapolated from Fig. 3C). They then employed a more precise method, intrinsic fluorescence, to assess ligand binding. However, due to technical limitations, they could only apply this technique to the isolated SBP domain, which appears to be a monomer. This approach yielded a much higher binding affinity with a K_d of 0.18 μM. While the thermal stability method may be less accurate because temperature increases can skew results, leading to higher K_d values, the significant 10,000-fold difference between the two techniques is difficult to reconcile. One may argue that the native full-length dimeric tri-domain protein – the form responsible for the biological function – is much less effective in binding glutamate than the isolated monomeric SBP domain.

We thank the reviewer for their comment and for their detailed evaluation of our data. However, we think there has been a misunderstanding. The Reviewer suggests that the DSF glutamate titration (Fig. 3C) was performed with the *full-length tri-domain* protein, but this is not the case. The DSF and the tryptophan fluorescence were *both* performed using only the isolated SBP domain. The differences observed in the apparent K_d from each technique referred to by the Reviewer stem from peculiarities and inaccuracies inherent in DSF. As the reviewer points out, using DSF to derive binding affinities can be very inaccurate. One of the primary issues is that for any particular protein, one cannot assume the K_d is constant at different temperatures, and as DSF is based on a temperature ramp, isothermal measurements are not possible using the standard protocol. Based on our previous DSF experience with other proteins and our data for the isolated SBP domain presented here, we always avoid using DSF to attempt to derive

K_ds, and use it only for qualitative screening. Fortunately, in this instance, intrinsic fluorescence titration was available to us, which is more precise and very widely used for SBP binding affinity analysis.

What is the model equation used to determine the K_d? This is not reported.

Apologies for this omission. We have now added the binding equation to the methods section.

The authors should investigate the binding mechanism more thoroughly, especially considering the novelty of the quaternary structure of the protein. The binding could involve allosteric or sequential mechanisms, which would require more complex models to accurately describe the binding process. which could be allosteric or sequential.

Our intrinsic fluorescence binding assays spanned a wide range of ligand concentrations from 25 nM to 1 μM and there was no evidence of sigmoidicity that would be characteristic of allostery. Therefore, for the isolated SBP, which is monomeric, we have provided strong evidence that it is a single site binding event, which is consistent with all other TRAP SBPs. The Reviewer's suggestion that the situation could be more complicated in the full-length protein is very interesting and will certainly be the subject of future follow-on studies. However, as our primary binding assay, intrinsic fluorescence titrations, is not compatible with the full-length protein (as detailed in the original manuscript), we cannot currently address this experimentally within a reasonable timeframe.

Additionally, to improve the clarity and reliability of the results, the authors should provide a fitting curve with error bars that reflect the variability of replicates. This would allow for a more accurate representation of the data and help assess the precision of the K_d determination. A representative curve without error bars can be misleading, especially when interpreting binding affinities from a single dataset.

We have now included all the binding curves used to determine the average K_ds reported in the manuscript (SI Fig 3).

The methods section could benefit from additional details to enhance reproducibility. For instance, the specific buffer used in the intrinsic fluorescence (ITF) assays should be clearly stated, as buffer conditions can significantly affect the outcome of binding studies. Including information such as buffer composition, pH, and ionic strength will allow others to replicate the experiments accurately.

Apologies for this omission. We have now included this information (50 mM Tris, pH 7.4, line 156). Thanks for spotting this.

Regarding the modeling tests with AlphaFold, more details on the parameter settings are essential. Specifically, the authors should mention:

The number of cycles used in the AlphaFold prediction process. This information is critical for understanding the level of convergence and whether additional cycles might improve accuracy.

Whether the highest scores (e.g., pLDDT or predicted local distance difference test scores) reached saturation. If the scores plateaued, this would indicate that further

iterations wouldn't improve the model's accuracy. Conversely, if the scores did not saturate, it could suggest that additional cycles could refine the model further. Given the availability of the AF3 code, we have repeated all the predictions with this version. In the case of the dimer and heterotetramer, we needed to extend the prediction up to 1500 models to reasonably converge the score values (see box plots in SI Figs. 5B and Fig. 6D). Instead, for the GltS dimer we needed to ramp the predictions up to 5000 (SI Fig. 6A-C). We added the details of each AlphaFold 3 prediction run in the methods.

Please also see our response to Reviewer #2's point (3) above.

Minor comments

Glutamate, along with the other potential ligands, should also be included in Figure 2C. We have added the chemical structures of the potential ligands and glutamate to SI Fig. 2. We tried adding them to the main figure as suggested by the reviewer, but we feel it cluttered the figure unnecessarily. Therefore, we have added it as SI.

Fig. 1A: choose a different color for the HD interdomain to better distinguish it from the adjacent SBP domain.

We thank the reviewer for the suggestion, but we would like to keep the colour scheme. It is colourblind compatible and we think it is sufficiently different.

Line 14. The Abstract starts with Tripartite ATP-dependent periplasmic (TRAP) transporter...an oversight about the ATP-dependence statement must be corrected. (see the discrepancy with line 45).

Corrected.

Line 41. A serial comma after "high affinity" might improve readability.

Thanks. Changed.

Line 68. "relatively poorly characterized" is slightly redundant and confusing; I suggest killing the adverb "relatively."

Adverb deleted.

Lines 71-74. The sentence encompassing lines 71-74 is long and could be split for readability.

This long sentence has been divided into two.

Line 79. A comma after "level" is necessary.

Comma added.

Line 83. "While there are examples of orphan TRAP SBPS..." add reference(s) for this sentence.

Reference added.

Line 373. It seems that the overlap of the two genes is by 4 bp. Please re-check.

We re-checked the overlap and we believe it is 3 bp. *gltS* (BP0402) ends on position 402157 and BP0403 begins on 402154, the difference being 3 bp. The diagram on

MicrobesOnline says 4 bp, but we believe this is an error based on the above information.

Line 377. A Reference must be provided for SeedViewer tool.
Thanks. Reference now included.

Line 540-542. To improve clarity and ensure the most up-to-date references are used, the outdated reference (Ref. 50, which is around 20 years old) should be replaced when introducing the Bordetella Uptake genes (Bug) of the tripartite tricarboxylate transporter (TTT) family in the discussion. Instead, use the more recent 2024 reference: DOI: 10.1002/2211-5463.13876. Since Ref. 50 is more relevant to the subsequent sentence; it should remain there where it is correctly cited. This ensures that the discussion is both current and appropriately referenced.
Huvent et al, 2006, has been replaced with Sorci et al, 2024, when first introducing the Bugs.